# MSC-DeepFM: OSM Road Type Prediction via Integrating Spatial Context Using DeepFM

Yijiang Zhao [1,2,3], Yahan Ning [1,2,3], Haodong Li [1,2,3], Zhuhua Liao [1,2,3], Yizhi Liu [1,2,3,*] and Feng Li [1]

[1] School of Computer Science and Engineering, Hunan University of Science and Technology, Xiangtan 411201, China; zhaoyj@hnust.edu.cn (Y.Z.); ningyahan@mail.hnust.edu.cn (Y.N.); johnlee97@foxmail.com (H.L.); zhliao@hnust.edu.cn (Z.L.); 1050023@hnust.edu.cn (F.L.)

[2] Hunan Key Laboratory for Service Computing and Novel Software Technology, Hunan University of Science and Technology, Xiangtan 411201, China

[3] Metaverse Innovation Research & Development Institute, Hunan University of Science and Technology, Xiangtan 411201, China

[*] Correspondence: yizhi_liu@sina.cn

**Abstract:** The quality of OpenStreetMap (OSM) has been widely concerned as a valuable source for monitoring some sustainable development goals (SDG) indicators. Improving its semantic quality is still challenging. As a kind of solution, road type prediction plays an important role. However, most existing algorithms show low accuracy, owing to data sparseness and inaccurate description. To address these problems, we propose a novel OSM road type prediction approach via integrating multiple spatial contexts with DeepFM, named MSC-DeepFM. A deep learning model DeepFM is used for dealing with data sparseness. Moreover, multiple spatial contexts (MSC), including the features of intersecting roads, surrounding buildings, and points of interest (POIs), are distilled to describe multiple types of road more accurately. The MSC combined with geometric features and restricted features are put into DeepFM, in which the low-order and high-order features fully interact. And a multivariate classifier OneVsRest is adopted to predict road types. Experiments on OSM show that the proposed model MSC-DeepFM achieves excellent performance and outperforms some state-of-the-art methods.

**Keywords:** road type prediction; multiple spatial contexts (MSC); DeepFM; OpenStreetMap; deep learning

## 1. Introduction

Volunteered Geographic Information (VGI) have been widely concerned by academia and industry since it was coined by Goodchild in 2007 [1]. OpenStreetMap (OSM) is one of the most successful VGI projects. According to OSM official statistics, it has more than 9 million registered users so far. Benefiting from numerous volunteers and their familiarity with surrounding features, geospatial data are updated on the OSM platform frequently and quickly. Now various applications are derived and expanded on OSM spatial data because the data is free for all uses [2]. And OSM is a potential source of geospatial open data for monitoring sustainable development goals (SDG) indicators [3]. Improving the quality of these crowdsourcing data has significant implications for monitoring and achieving SDGs, such as zero hunger, sustainable cities, ensuring tenure security, and preserving biodiversity [4].

Nevertheless, there are some issues with spatial data quality because the OSM platform does not have a rigorous error detection and notification mechanism during the contributors' submission process and many contributors lack knowledge related to geography and geographic mapping, such as inaccuracy and incompleteness, and data quality vary with different countries and regions [5]. And the contributing experience and skill of contributors also vary greatly. Therefore, it is very difficult for most of them to describe OSM geographic

elements with accurate semantic attributes (tags in OSM), which leads to certain semantic quality problems in the OSM dataset, especially for some geographic objects with similar types [6]. To some extent, these issues have hindered the development of the OSM and reduced its role as a valuable source for monitoring some SDG indicators. Therefore, the research on how to improve the quality of OSM data [7–9] has been a popular topic in the academic community in recent years.

Spatial data mainly contain three basic features: spatial, thematic, and temporal features. The thematic features of geographic elements in OSM are mainly described by tags. It can be divided into element class tags and other attribute tags [6]. Element class tags are applied to differentiate from other types of geographical elements, such as highways, major roads, or residential roads. These tags are very important attributes that connect OSM elements and map layers [6]. Other attribute tags describe other characteristics of the OSM elements, such as road name, width, and speed limit.

The geographical information quality of OSM has been extensively studied recently. Most of them have focused on two aspects: quality evaluation [10–12] and quality improvement [7,13]. In terms of quality improvement, many scholars pay more attention to tag recommendations of OSM geographic objects [9,14–22]. Existing tag recommendation methods are mostly based on the characteristics of OSM elements themselves [14–19]. For example, Storandt et al. proposed a system to recommend suitable road labels only according to the name of points of interest (POIs) [15].

The road is one of the most important elements in OSM, and it is the basis of numerous applications such as navigation and network analysis [23–26]. Therefore, the tag recommendation of the OSM road network is of particular importance. At present, most of the research on road tag recommendation considers the object's characteristics and its restricted characteristics [18–20]. In addition, the experimental data for the studies are generally selected from the areas with rich OSM data (such as London, UK), and the OSM data of these places are recognized to be extremely useful [27]. In other words, these models are generally only effective in handling dense OSM datasets. However, insufficient quality and availability of the OSM data relatively in some economically underdeveloped countries and regions often limit their application. The problems of incomplete geographical objects or inaccurate semantic descriptions of geographical objects exist in various degrees. Thus, it is still challenging to further improve the accuracy of tag recommendation.

According to Tobler's First Law of Geography [28], the spatial distribution of geographical things or attributes is interrelated with each other, and it appears with clustering, random, and regular characteristics. The relationship can be described by the spatial context. Hence, spatial context extraction from the surrounding environment of geographic objects can enrich their characteristics and is very useful for predicting their types. As shown in Figure 1, it is easy to find out the difference among the three road types in the OSM platform: secondary, tertiary, and residential roads. The tertiary and residential roads offer more opportunities to be close to residential buildings (regularly arranged buildings in the figure), while secondary roads have only a small part adjacent to residential houses. Its main function is to connect specific administrative centers, traffic hubs, commercial zones, etc., of which characteristics are straight and spacious.

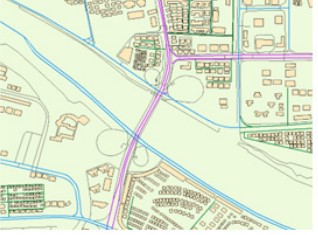 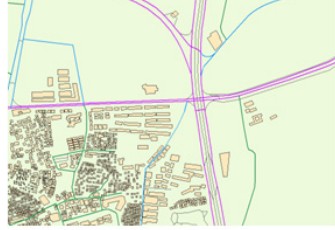

**Figure 1.** An example of different road types in OSM, where secondary, tertiary, and residential roads are purple, blue, and green, respectively, other types of roads are black, and buildings are yellow rectangles.

Several studies have shown that it is feasible to extract spatial contexts (SC) and apply them to tag recommendation of spatial objects [9,20–22], which can improve the recommendation accuracy in the case of poor data integrity and low semantic accuracy in OSM regions. Ali et al. analyzed the spatial context of fuzzy grassland classification tag recommendation and counted the relevant regional entities, such as "amenities" and "leisure" and some linear entities in the research regions [21], in which the effectiveness of spatial context deployment has been verified. However, they did not adequately consider the influence of the surrounding objects, for example, POIs around the park regions. Alghanim et al. added the building context into the feature matrix of road elements, and they used the random forest (RF) algorithm to recommend road tags [6]. The work achieved good results on their dataset, but they ignored the influence of other spatial contexts, such as connecting road context, Zhao and Tang proposed a system to recommend suitable building labels that introduced external semantic features, including the location features of buildings, spatial co-location patterns of points of interest (POI), nighttime light, and land use information of the buildings [22].

In general, most of the current tag recommendation methods rely on the semantic and geometric features of the elements themselves or focus on partial spatial context, which makes it difficult to accurately describe the geographic objects especially when the data are sparse and there is a problem of semantic completeness. It will lead to a decrease in recommendation accuracy and poor recommendation performance for road tags. Therefore, we focus on the roads of OSM, explore the influence of the spatial context of the different levels of roads, and propose a road type prediction model considering multiple spatial features, geometric features, and restricted features. The main contributions of this paper are as follows:

(1) Motivated by Tobler's First Law, we propose a representation method of multiple spatial contexts (MSC) which include the features of intersecting roads, surrounding buildings, and POIs, which are distilled to describe different road objects in OSM more accurately. Four kinds of road topological relations are used to model different kinds of intersections. To the best of our knowledge, we are the first to consider road topological relations and the POI context of road objects for road type prediction.

(2) We propose a novel framework named MSC-DeepFM for OSM road type prediction to cope with data sparseness. It fuses MSC, geometric features, and restricted features of OSM geographic objects into the DeepFM model. Thus, the framework can deal with data sparsity and the lack of semantic integrity through full interactions of multi-class features. Moreover, the OneVsRest algorithm was devised to extend the model to multiple classifiers.

(3) We design a representation method for the eigenmatrix of the spatial context, restricted features, and geometric features. We consider the spatial context features of road elements from different sources, such as the topology structure of road elements, surrounding buildings, and POIs. After combining them with the restricted features and geometric features of the roads, a total 62-dimension vector is constructed. We also design an approach that can automatically obtain the spatial context around the road.

(4) The OSM Guangdong data are used for our experimental evaluation. The F1 score of our proposed model is 93.27%, which outperforms the existing tag recommendation model for OSM in the literature [9] and many state-of-the-art machine learning methods. Furthermore, the prediction accuracy of our model is obviously improved after integrating the features of intersecting roads.

## 2. Related Work

The related work in this research mainly focuses on quality evaluation and improvement of OSM data.

### 2.1. Quality Evaluation of OSM Data

In recent years, the ability of novice contributors to accurately describe spatial geographic data poses many concerns for OSM data quality [8]. Therefore, many studies

have focused on evaluating the accuracy and completeness of OSM data. International Cartographic Association (ICA) [28] developed seven rules for assessing the quality of spatial geographic data. Based on these seven rules [29], Barron et al. extended the rules, including the semantic, geometric accuracy, and availability of spatial data. Moreover, since contributors are an important part of map production, data-centric and contributor-centric assessments are often combined [11,12,30–33]. Overall, the OSM quality assessment can be divided into extrinsic quality measures and intrinsic quality measures.

Extrinsic quality measures: This type of research compares OSM data with authoritative data from other official institutions for external evaluation [10,34–36]. Such methods rely on external data. However, authoritative datasets are more difficult to obtain than public ones, and the update efficiency of their data is sometimes slower than the fast-changing OSM data. All these factors constrain such evaluation methods in applications and expansions.

Intrinsic quality measures: Intrinsic method does not rely on external or authoritative data sources for validation [37,38]. The methods can measure the accuracy of OSM data by assessing changes in historical versions of the data, or by associating user reputation. As an example, Fogliaroni et al. calculated the quality score of geographic features by analyzing the geometric, qualitative, and semantic changes in the edited version of history. They used it to approximate the quality score of spatial data at the end [8]. Zhou and Zhao used spatial similarity and geometrical similarity to calculate the similarities between the versions. The reputation of the contributor was obtained by analyzing the complicit assessments computed by version similarity [28]. Mullagann et al. analyzed the spatial semantic relations of point features. The spatial semantic interaction was used to measure the semantic similarity of the change history of geographic elements [33].

### 2.2. Quality Improvement of OSM Data

The quality improvement of OSM data has received much attention from many researchers. It can be divided into two aspects: identifying and correcting erroneous data for OSM, and OSM tag recommendation.

Identifying and correcting erroneous data for OSM: The early literature on improving the quality of crowd-sourced geographic data focused on detecting and modifying error elements. For example, Vargas et al. used a Markov random field method to maximize the correlation among annotations of OSM buildings and predicted building probability maps. After removing several redundant geometric annotations through the relationship between building probability maps and thresholds, they used CNN to predict and add new architectural geometric annotations [7]. Kashian et al. analyzed the "semantics" of the newly contributed data by identifying potential patterns of coexistence between POIs and other geographic features. They calculated the likelihood of a POI being registered at this location to improve the detection and verification system of the OSM platform. The location accuracy of registered POI in OSM can be improved [13]. These studies have contributed significantly to improving the semantic quality of OSM databases.

OSM tag recommendation: OSM does not have a proper tag verification mechanism, which leads to a problem in that the OSM tags vary greatly with different contributors. A tag recommendation method is a good method for this issue and can significantly improve the quality of OSM data [8]. There have been many studies on OSM and other open data in recent years. For example, Arnaud et al. established a tag recommendation system named "OSMantic". By calculating the corresponding semantic similarity score, the system, which gives timely relevant suggestion tags by calculating the corresponding semantic similarity score when users submit commit them. And the system will give some semantic accuracy hints when the score is too low [14]. Storandt et al. developed a recommendation system that only needs the POI name to recommend appropriate tags [15]. Jilani et al. constructed the features of road elements, such as degree distribution, intermediary centrality, node number in a bounding box, etc. The model constructs and represents the road and its features by using a graph structure. They used an artificial neural network (ANN) to train

the model and recommend tags [16]. Corcoran et al. focus on geometric features and define a series of geometric features about road elements, such as degree, road curvature, parallelism, etc. Finally, their model reported 68% and 65% weighted accuracy and recall values, respectively [18]. Hacar used geometric and semantic features to classify and recommend the leisure tags of polygons [19]. Tag recommendation can motivate contributors to contribute correct tags which are highly effective in improving quality. Therefore, it is currently a widely studied method. However, these studies only consider geometric or other semantic features of the element itself, which mostly depend on the quality of the OSM data, and it often struggles to adapt to semantically incomplete datasets.

For geographical spatial features, each element is related to other similar elements, and only the distance determines the size of the influence [27]. In the current research, there is a tendency to combine features of geographic elements with the spatial context, and there are many achievements. For example, Zhang et al. used geometric features and restricted features of road elements to detect label tag semantic inconsistency and other problems, while giving intelligent suggestions based on the information available in the spatial context of the problem data [20]. Alghanim et al. used building context as a feature to analyze road elements and used a 20 M to 200 M linear buffer to count context semantics. They developed classifiers to recommend classification tags for road elements based on this method [9]. Ali et al. identified and predicted several grassland fuzzy categories based on contextual attributes and topological features by analyzing the case of building elements and road elements versus object elements with three selected topologies. Among them, the connecting road context applies several road element categories related to park grass, including "foot" and "bike" [21].

## 3. Method

This section begins with the overall workflow and methodological discussion of the paper. It mainly includes the overall workflow, feature selection, and our road prediction model. The model section is devoted to feature matrix construction, OneVsRest strategy, and algorithmic pseudocode.

### 3.1. Overall Workflow

To effectively recommend OSM road object element class tags, this paper proposes a method based on the MSC-DeepFM model. The research framework in this paper mainly consists of three steps: data preprocessing, feature engineering, and road type prediction. This paper mainly focuses on the latter two parts. In summary, we perform some experiments based on the MSC-DeepFM model to efficiently recommend class tags of OSM road object elements. The workflow is illustrated in Figure 2. In the first part, the experimental data mainly includes road data, building data, and points of interest (POI) data extracted from OSM. Data processing mainly consists of analyzing and processing the semantics of POI and building data in the OSM, as well as grading and screening road objects. In the second part, feature extraction is mainly concerned with constructing feature vectors including geometric features, restricted features, and spatial context features, in particular, constructing spatial context features. Meanwhile, we perform feature fusion on the constructed features. In the model construction and prediction part, the constructed features are mainly fed into DeepFM. Then, we use the constructed OneVsRest classifier to make multi-classification predictions for road tags. Finally, it will generate a list of tag recommendation probabilities for the user.

### 3.2. Feature Selection
3.2.1. Spatial Context Features

Since there is a certain connection between geographical objects and their surrounding geographic objects [27], many scholars have made use of this connection to conduct research, such as Ali et al., who used three topological relations in the nine-intersection model [21] to describe the relationship of spatial context related to grassland. After that, they constructed

a classifier to give intelligent suggestions for fuzzy grassland-type tags. Amerah et al. analyzed the value of houses by counting the types of streets around houses [6]. Hence, this paper considers the spatial context of the roads in the study of OSM road type prediction to improve the prediction accuracy. The spatial context mainly considered in this paper includes roads, buildings, and POIs.

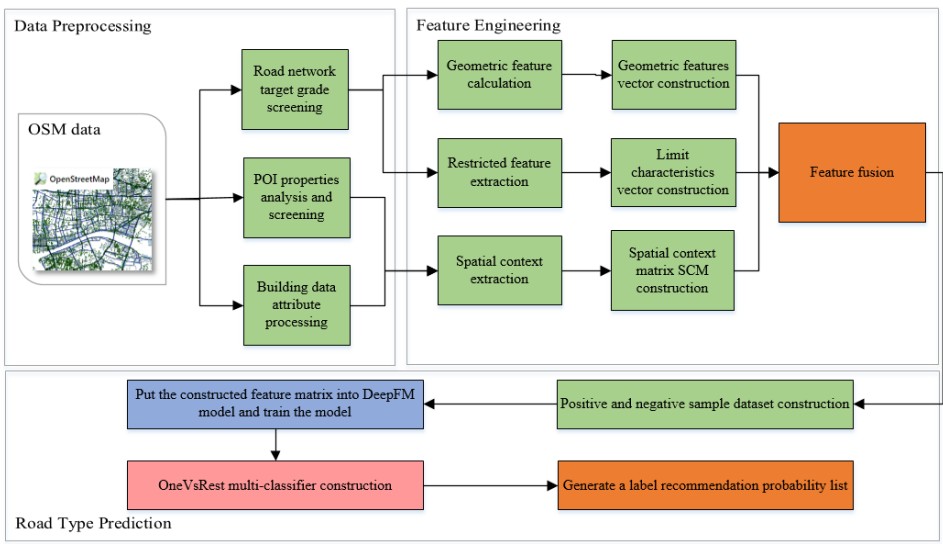

**Figure 2.** The framework of MSC-DeepFM.

Connecting road context: The road network consists of road junctions of different levels, which often hide some semantic information such as road type. For example, in the experimental dataset, the road type is tertiary, and the largest proportion of roads intersecting with it is residential, accounting for 23.38%. Because the tertiary highways of China are three-level highways, which is a type of low-level road. The primary function of a tertiary road is to connect local towns and villages, remote suburbs, or function areas, and most tertiary highways can be connected with residential roads.

There are 14 kinds of spatial topological relations between two independent straight lines [39]. Due to the particularity of road type prediction, connecting roads play an important role in road type prediction. Therefore, we only consider the relations of the intersection roads in this paper and propose four kinds of line topological relations for modeling. As shown in Figure 3, the spatial context of a road is obtained by combining the four relations of connecting roads. At the same time, the type of connecting roads is also considered in the feature construction of our model.

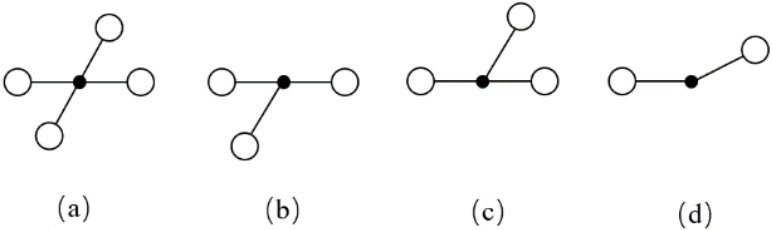

**Figure 3.** Four lines topological relations, where (**a**–**d**) represents crossing relation, merging relation, outflowing relation, and junction relation, respectively.

Building context: The building context refers to the distribution of various types of buildings within a certain buffer of the OSM object. By establishing a road buffer and counting the types of buildings in the vicinity of the road, dozens of building type tag semantics, such as houses, apartments, and businesses, are screened as building context information based on the value of the surrounding building semantics. According to

the research, there are a large number of buildings in the 20 M buffer zone of residential roads [9], and most buildings tend to be residential, commercial, etc. For high-grade roads, the spatial context has a larger influence range than for low-grade roads. The buffer zone was set to approximately 200 M for data analysis and comparison. The example buffer graph is shown in Figure 4, where the blue polygon is the building element, which contains the type semantics.

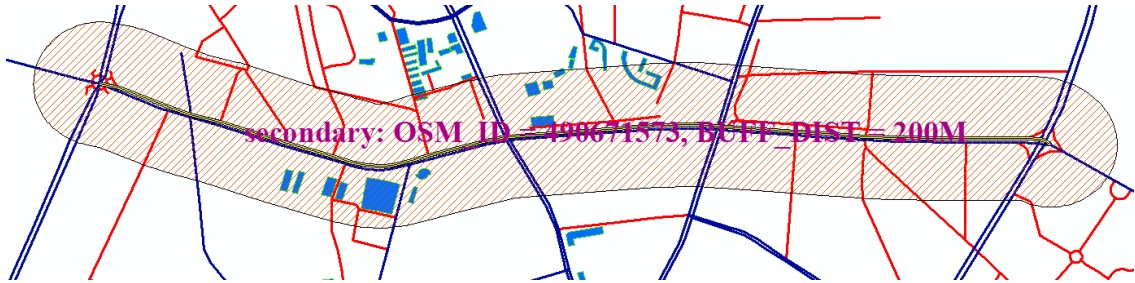

**Figure 4.** The 200 m buffer of a secondary road whose OSM ID is 490671573. The red line represents the low-grade roads, and the blue line represents the major roads. The buffer is shaded in red and centered on the object road, which is marked with three yellow lines.

POI context: The POI context is the distribution of various types of POIs in the buffer within a certain range of the OSM object. OSM elements consist of points, lines, and planes, and POI is the point element. In GIS, a POI can be a house, a business, a mailbox, a bus stop, etc. The semantics of these POI points are closely related to the surrounding roads. For example, a commercial street road may have a bus stop or subway station entrance and underpass next to it. These elements are closely related to the road and affect its level. In the experiment of this paper, given that the object element is roads, the POI points are selected for semantic categories related to road traffic, including "bus_stop", "crossing", "traffic_signals", "motorway_junction", and "fuel". They are bus stops, pedestrian walkways, traffic lights, motorway connections, and fuel stations; fuel stations are an obvious feature of a highway. On all highways, there are service stations for fuel, so they can characterize the highway.

### 3.2.2. Geometric Features

Geometric features refer to features such as the degree of bending and the number of junctions reflected by the shape of the road. Different grades of roads have different characteristics in terms of road length and bending degree [18]. In China's national standard GB/T 920-2002 [40] road grade regulations, secondary roads can be described as straight and spacious roads, while tertiary roads can be described as low-speed limits. The route fluctuates or turns with the terrain. At the same time, a large amount of data shows that most of the low-grade roads have a higher degree of bending than the high-grade roads. Therefore, the geometrical features of roads in this study mainly include the road length, the number of road nodes, the road curvature, the number of intersections, and the number of dead ends. The last three features are described as follows:

Road curvature: It is calculated by the ratio of road length to the number of nodes. The smaller the ratio, the more curved the road will be.

The number of intersections: It refers to the number of intersections contained in a road section. when the road length is the same, the more intersections there are, the lower the road grade tends to be.

Dead end: It is one of the more common features of residential roads.

### 3.2.3. Restricted Features

Restrictive rules are features of road traffic rules or related features that impose certain constraints on traffic behavior, such as maximum speed limits, one-way streets, etc. These

road rules constitute a fundamental property of roads and are an important feature of them. In this paper, we describe their intrinsic properties for roads. In this paper, five types of restricted features are considered. The tags in the OSM are "oneway", "maxspeed", "layer", "bridge", and "tunnel". Their meaning is described as Table 1.

**Table 1.** Tags in the OSM.

| Tag | Description |
| --- | --- |
| Maxspeed | The maximum driving speed of the road. |
| Oneway | Whether the road is a one-way street. |
| Layer | The level of the road on the ground. |
| Bridge | Whether the road includes a section with a bridge. |
| Tunnel | Whether the road contains a tunnel section. |

### 3.3. DeepFM Model

DeepFM is a CTR prediction neural network based on a factorization machine proposed in 2017 [41]. DeepFM is a framework that integrates the "Wide" and "Deep" model for joint training. By comprehensively utilizing the memory ability of the shallow model and the generalization ability of the deep model, the accuracy and extensibility of the recommendation system can be taken into account by a single model. It models low-order feature interactions like FM and models high-order feature interactions like DNN, which makes up for the defect that some linear models cannot learn feature interaction. As shown in Figure 5, DeepFM includes factorizer FM and deep neural network DNN, which share the embedding output vector from the embedding layer. The output formula of the model is as follows:

$$\hat{y} = \text{sigmoid}\left(y_{FM} + y_{DNN}\right) \tag{1}$$

where $\hat{y}$ is the prediction result, $y_{FM}$ and $y_{DNN}$ are the output results of FM and DNN, respectively, and sigmoid is the activation function. The output layer accumulates the results of the FM layer and the hidden layer, combining low-order and high-order feature interactions. It then undergoes a non-linear transformation to obtain the predicted probability output.

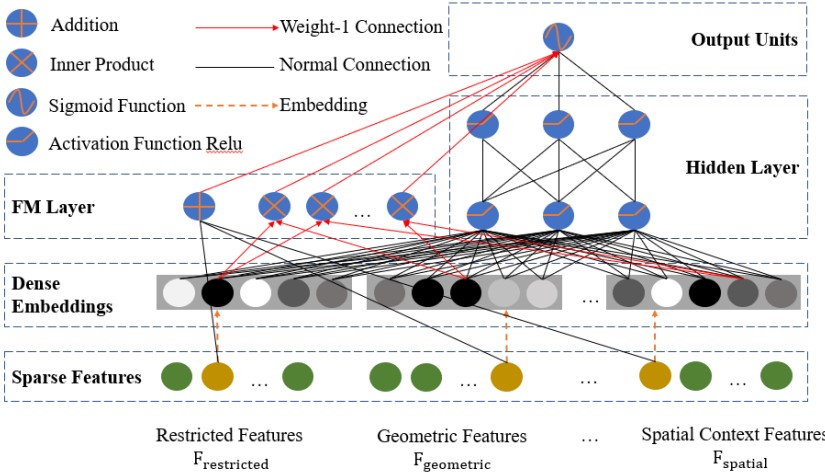

**Figure 5.** MSC-DeepFM model structure diagram.

Before the data is input into the model, they will go through an embedding layer. For sparse dimension vectors, it will be embedded into low-dimensional dense vectors. It is worth noting that no matter how many dimensions of input vectors, their embedding output is the quantity of fixed dimension.

In addition, the Sparse Feature and the Dense Feature are included in the input vector. Among these features, the implicit vector $V_i$ and the scalar $W_i$ are trained for a feature i. For the feature i, the scalar $W_i$ is used to measure its first-order importance, while the implicit vector $V_i$ is used to measure its interaction with other features. $V_i$ will then be fed into the FM component to model the second-order feature interaction. Since the Deep part shares input with the FM part, $V_i$ will also feed into the deep component to model higher-order feature interactions. Meanwhile, for all parameters, including $W$, $V_i$, and network parameters, the following $W^{(1)}$ and $b^{(1)}$ will be jointly trained for the combined prediction model. The output formula of the embedded layer is as follows:

$$a^{(0)} = [e_1, e_2, \ldots, e_m] \tag{2}$$

where $e_i$ is the embedding of the ith feature and m is the number of features. The embedding vector is then fed into DNN and FM.

Theoretically, as long as the number of hidden layers is sufficient, DNN can fit any function. Therefore, the model inputs the shared implicit vector $V_i$ to DNN as a weight parameter and performs higher-order interaction with it. In DNN, the positive feedback process of the model is as follows:

$$a^{(l+1)} = \sigma\left(W^{(l)}a^{(l)} + b^{(l)}\right)\#(1) \tag{3}$$

where l is the number of layers, σ is the activation function, and Relu is used as the activation function for the hidden layer. $W^{(l)}$, $a^{(l)}$, $b^{(l)}$ are the lTH layer model weight, output, and bias terms, respectively. Then, it will produce a dense solid values eigenvector. Finally, the model will input it into the activation function of road grade classification prediction: $y_{DNN} = \sigma\left(W^{|H|+1} \cdot a^H + b^{|H|+1}\right)$, including $|H|$ is the number of layers of hidden layers.

FM is a second-order polynomial regression model with multiple feature interaction improved by the linear regression model. Because of the introduction of the second-order interaction term, the model can learn the second-order interaction features except for the first-order features. The model output formula is as follows:

$$y_{FM} = < W_i, \ x > + \sum_{j_1=1}^{d} \sum_{j_2=j_1+1}^{d} < V_i, V_j > x_{j_1} \cdot x_{j_2} \tag{4}$$

where $w \in R^d$ and $V_i \in R^k$. The first half of the $< W_i, \ x >$; the unit maps the first-order feature; and the inner product unit maps the second-order feature interaction.

### 3.4. Feature Matrix Generation for DeepFM

Before feature input into DeepFM, each feature should be encoded and combined to form a feature matrix, and then normalized.

(1) Construction of spatial context matrix: Firstly, we extracted the roads by classification and traverse the road. The road semantics intersecting the road in geometric topology is extracted as the connecting road context. The vector $R_i = [r_1, r_2, \ldots, r_n]$ is taken as the connecting road context representation of the object road, where $r_i$ is the semantic representation of the road intersecting the object road, and n is the total number of road semantic types in the data. Then, a threshold of 200 M [9] is set to extract the building semantics intersecting with the road buffer threshold, and the vector $B_i = [b_1, b_2, \ldots, b_m]$ represents the building context semantics of the object road, where $b_i$ represents the building semantics intersecting with the object road buffer, and m represents the total number of building semantic types in the data. For the POI context, a buffer with a threshold of 5 M is set, and the road is extracted in recycling. The vector $P_i = [p_1, p_2, \ldots, p_k]$ is used as the POI context semantic representation of the object road, where $p_i$ is the POI semantic type whose threshold is set with the object road, and k is the total number of POI semantic types

in the data. These three vectors will be used as the SC of the road object. Finally, these three semantic vectors are described by a spatial context matrix (SCM).

(2) Construction of other feature vectors: Other features mainly include geometric features and restricted features. After traversing the object elements and calculating the geometry, the geometric features are expressed as geometric feature vectors of the road by using vector $G_i = [g_1, g_2, \ldots, g_l]$, which has a size of is $1 \times 3$ dimensions. Restricted feature construction, including five features "oneway", "maxspeed", "layer", "bridge", and "tunnel". Similarly, the vector $Rst_i = [r_1, r_2, \ldots, r_o]$ is expressed as the restricted feature of the road, where the vector size is $1 \times 5$ dimensions.

(3) Recommendation model fusion: The above spatial context, geometric features, and restricted features are combined into a multi-dimensional feature vector matrix, and they are fused into the DeepFM model in the way of feature engineering. DeepFM is a fusion method of deep learning DNN and factorization machine FM, which combines a recommendation algorithm with deep learning so that training and learning of DeepFM can be carried out in the form of interaction between low-order features and high-order features. The spatial context matrix, combined with the geometric feature matrix and restricted feature matrix, is put into DeepFM to learn the relationship between road types and spatial context. After the input feature matrix is trained, the OneVsRest algorithm is used to obtain the predicted road type probability list. The model will recommend the TOP1 road type as the final prediction result, which can provide OSM contributors with references.

As shown in Figure 6, the spatial context matrix is composed of three parts, all of which describe the spatial context using statistical semantics and embedding matrices. The matrix of context part is called SCM (spatial context matrix) in this paper. The whole input vector matrix is constructed by fusing spatial features, restricted features, and geometric features. The overall construction is shown in Figure 6, where B1, B2, B3, and so on represent building context semantic tags, such as "type = residential". Similarly, R1, R2, R3, and so on represent connecting road context semantic tags, and P1, P2, and P3 represent POI context semantic tags.

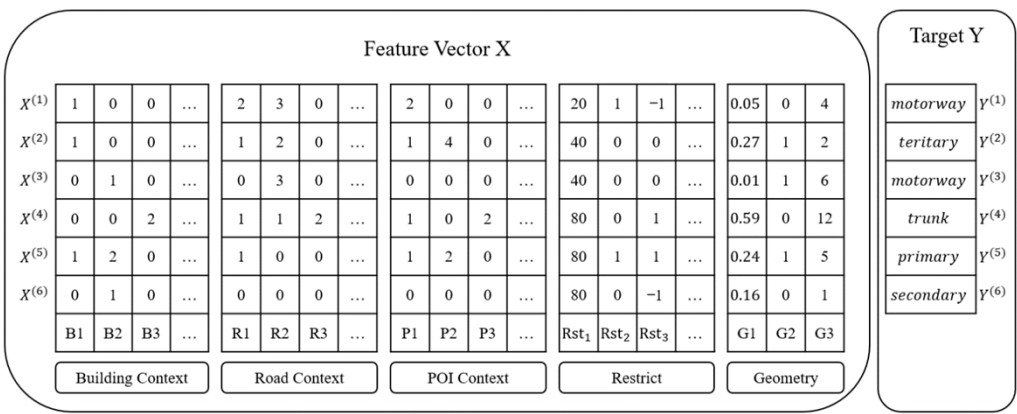

**Figure 6.** Feature eigenmatrix integrating with multiple spatial features.

### 3.5. OneVsRest Multi-classification Policy Algorithm

OneVsRest transforms the binary classification algorithm into a multi-classification algorithm through multiple iterations and pairwise comparison, similar to OneVsOne. After a comprehensive comparison, this paper adopts OneVsRest's algorithm for the multi-classification prediction of road types. In the OneVsRest strategy, assuming n classification tasks, n binomial classifiers will be established. Each classifier takes one of the categories as a positive class and the other classes as a negative class. This model is denoted as $h_\theta(x)$, see Equation (5), where y represents classification category $i = (1, 2, 3 \ldots n)$. For prediction, the n binomial classifiers are used for algorithm classification, and the probability of data belonging to the current positive class is obtained. Finally, the positive class of the classifier

with the largest probability is selected as the final prediction result, namely $\max\left(h_\theta^{(i)}(x)\right)$.

$$h_\theta^{(i)}(x) = p(y = i|x;\theta) \tag{5}$$

## 4. Dataset Preparation

In this study, we focus on the road network data in OSM. The OSM data of Guangdong Province in China, which has a well-developed road network, was selected for our experiments. As the province with the highest GDP in China, Guangdong Province has seen the fastest economic growth and rapid growth in road laying. Its highway mileage ranks first in China and also when compared with the London dataset [6], which is recognized as the highest quality dataset. The Guangdong Province data has a certain semantic sparsity, which is helpful in testing the processing ability of the model on sparse data. According to the statistics, the OSM Guangdong dataset adopted in this experiment has a total number of 396,393 road samples, with a large proportion of road types including residential, unclassified, tertiary, and service roads accounting for 24.27%, 11.69%, 11.50%, and 11.41%, respectively. The lowest percentages include bridleway, track_grade2, track_grade5, track_grade3, and track_grade4, which are some of the more minor categories. In addition, we also used the building and POI data of Guangdong Province. According to statistics, the semantic uninformed rate of buildings is 61.78%, which shows the data is sparse to some extent. Building data contains a total of 68 semantic types, accounting for 142,355 data items. For POI data, due to the sparse problem, we selected five categories with relatively concentrated, which are all related to traffic, including "bus_stop", "crossing", "traffic_signals", "motorway_junction", and "fuel", with a total of 56,645 data items after processing. Due to some data accuracy problems of roads with finer grades, we selected major roads in Guangdong, China for our research, which includes five road types, i.e., motorway, trunk, primary, secondary, and tertiary, with a total of 121,970 data items. The details of these road types are listed in Table 2.

**Table 2.** Statistics of main roads in Guangdong, China.

| Code | Type | Description | Frequency | Proportion |
|------|------|-------------|-----------|------------|
| 5111 | motorway | Motorway/freeway | 22,073 | 18.09% |
| 5112 | trunk | Important roads, typically divided. | 18,591 | 15.24% |
| 5113 | primary | Primary roads, typically national. | 24,692 | 20.24% |
| 5114 | secondary | Secondary roads, typically regional. | 45,568 | 37.36% |
| 5115 | tertiary | Tertiary roads, typically local. | 11,046 | 9.06% |

Data preprocessing is a crucial step in facilitating accurate road type predictions. Specifically, spatial context features, geometric features, and restricted features are extracted for all the roads of the five types. For the spatial context features, POIs and buildings within a specified distance from the roads are extracted. Then, the spatial distribution features of POIs are analyzed, and the distances to the nearest roads are computed. According to the predetermined classification criteria, these POIs are classified, and the semantic features of the buildings around the roads are generated. Additionally, connecting road context is extracted through the topological relationships of the roads. And following that, the coordinates and restricted features of the roads are extracted from OSM data. The geometric features are computed using the coordinates of the roads. The corresponding feature matrix is constructed after removing data outliers and normalizing the data, serving as the input for the DeepFM model.

## 5. Experiment and Evaluation

The main objective of the experimental evaluation is to evaluate the modeling effectiveness of MSC-DeepFM. Several comparative experiments are conducted to analyze

the effectiveness of the classification model and the proposed features. In addition, the ratio of the training set to the test set is 7:3, and the experiment has achieved the best results for DeepFM parameter tuning. The machine learning models used in this work are evaluated using standard evaluation metrics, namely Accuracy, Precision, Recall, and F1 score. Ten-fold cross-validation was used in all experiments and we took the ten-fold mean of the results as the final experimental result, which brought the results closer to the actual performance.

The baseline models used in our experiments for comparison are random forest [6], AdaBoost [42], KNN [43], FNN [44], and DCN [45]. Among them, random forest is a Bagging algorithm based on the decision tree, which is different in that random attribute selection (subsampling on features) is also added in the training process of the decision tree. In contrast to decision trees, random forest is not prone to overfitting and it is robust. While Adaboost is a strong classifier generated by combining weak classifiers, which is an ensemble learning algorithm. Its result is a fusion recommendation of multiple weak classifiers, which can reduce the impact of some models with poor classification results. FNN, DCN, and DeepFM have combined DNN Deep neural networks. Those algorithms are click-through-rate (CTR) prediction models based on wide and deep structures. FNN is a feed-forward neural network that requires FM pre-training. The embedding parameters may be over-affected by the FM. Furthermore, fuzzy neural networks only capture higher-order feature interactions.

In contrast, DeepFM does not require prior training and can learn both higher-order and lower-order feature interactions. DCN is a modified algorithm for FM, which focuses only on the interactions of second-order features but fails to capture higher-order features. The core of DCN is that it introduces the cross network to model higher-order features. It is worth noting that the vectors in the cross-network structure of DCN are all column vectors. The dimensions of its training parameters w and b at each layer are consistent with the dimensions of the output vectors, however, this training parameter is a very small order of magnitude. This also suggests that there is a limit to its expressive power. KNN is a K-nearest neighbor classification algorithm that is a non-parametric trained model, and it does not make any assumptions about the data. The algorithm divides the other categories according to the k nearest neighbors of the target, while it is insensitive to outliers and has fast training speed and superior performance.

The experimental evaluation results are presented in Table 3, where we compared the performance metrics of various machine learning algorithms. The experimental data are Guangdong roads, and the five road types collected from the data are used as samples. All the feature metrics, including restricted features, geometric features, and spatial context features, are used. Based on the comparison experiments, we adopted the best parameter settings. The model performance is compared with AdaBoost, FNN, KNN, and DCN. According to the results, the proposed OSM road classification tag recommendation model based on MSC-DeepFM outperforms other popular models in terms of Accuracy, Precision, Recall, and F1 score.

**Table 3.** The model performance compared with other baselines.

| Algorithm | Accuracy | Precision | Recall | F1 Score |
|---|---|---|---|---|
| Random Forest [9] | 29.70% | 31.17% | 38.63% | 48.49% |
| AdaBoost [42] | 82.24% | 74.47% | 81.78% | 84.10% |
| KNN [43] | 91.28% | 91.54% | 91.65% | 91.58% |
| FNN [44] | 89.89% | 91.84% | 88.95% | 90.26% |
| DCN [45] | 91.07% | 91.98% | 90.76% | 91.32% |
| MSC-DeepFM | 92.98% | 94.05% | 92.52% | 93.27% |

For KNN, the F1 score of our model increases from 91.58% to 93.27%, showing a significant improvement. Due to the low completeness and quality of the experimental data, the F1 score of the traditional model, AdaBoost and random forest, is 84.10% and 48.49%, respectively. Alghanim et al. used OSM London data to model roads in a building context [9]. The F1 score reached by the model is 83.78%. However, the ratio of the uninformed data types and insufficient quality accounted for more than 60%. The quality of the experimental dataset is far below that of the data set in London, UK. In this case, AdaBoost and random forest struggle to handle high-dimensional and sparse input data, resulting in poor performance and poor classification results. On the other hand, both FNN and KNN are deep models and their F1 score of results are 90.26% and 91.32%, respectively. Because of the limitations of FM, FNN is unable to play the role of deep models and performs relatively poorly. DCN and DeepFM are structurally parallel, that is, the cross network structure is trained jointly with the DNN. Nevertheless, in terms of input sharing, the two parts of the DCN are trained separately and output together at the end. In contrast, DeepFM shares the embedding results and feeds the hidden vectors to the DNN for higher-order interaction modeling, which can better learn the underlying feature relationships. Deep Neural Networks in DeepFM are used to learn the deep interaction relations between features and establish feature interactions between the spatial context of geographic object elements. These structures allow the model to perform well in the final tag recommendation, and it can handle sparsity better than the general decision tree models and other deep learning models. Second, the DNN part of DeepFM can perform deep learning on large samples, and its learning ability is better than that of general machine learning models. Moreover, it has good portability and can be adapted to most spatial datasets. When DeepFM is applied to OSM data with low completeness of buildings and POIs, the performance will decrease relatively. However, in the same low-completeness data, our model will also perform better than other popular models. We will deploy some relevant experiments in subsequent studies to verify this.

The confusion matrix results of MSC-DeepFM on the test set are shown in Figure 7, where the motorway and trunk have the highest accuracy of 1.0 and 0.97, respectively. For secondary roads, the tertiary misjudgment rate is the highest (0.11). According to the national standard GB/T 920-2002, the tertiary may be an arterial highway serving a county or town and may be connected to civilian roads and other interchanges, and it can be unsealed and has pedestrian zebra crossings. Secondary roads are also widely used in urban or suburban areas and secondary roads or as side roads. In general, only high-quality secondary roads can be incorporated into a high-grade highway without isolation or closure. There are a large number of planar intersections, which can easily be confusing to the tertiary. In addition, it is suitable for hilly areas with large terrain undulations. In many economically underdeveloped areas, secondary roads are widely used to replace primary roads. According to the statistics in Table 2, the frequency of secondary roads accounts is 37.36%. These causes can lead to a miscalculation between secondary and tertiary.

Comparison experiments: To compare the effect of different inputs of the model on the experimental results, a series of comparison experiments are conducted, which is shown in Figure 8.

In the experiments, different feature constructions are truncated to generate the input vectors. After feeding the restricted features into the model alone, the F1 score of the model is 86.21%. While inputting the feature vector composed of the restricted feature and geometric feature into the model, the F1 score is 86.86%, and it has a certain promotion. Nevertheless, when the restricted feature and spatial context (SC) are united as input, the model performance has a large improvement with an F1 score of 91.32%. It shows that spatial context played an important role in road type prediction. On the other hand, SC is composed of three parts of the context. The accuracy of the prediction is related to the richness of the context in model prediction. Finally, fusing the spatial context feature matrix SCM with restricted features and geometric features input into the model increases the F1 score to 92.48%. It proves that spatial context has a significant effect on model feature

interactions. As an additional feature description of the road, the spatial context implies the feature relation of the mutual influence of all objects in the geography, which performs well in the DeepFM model.

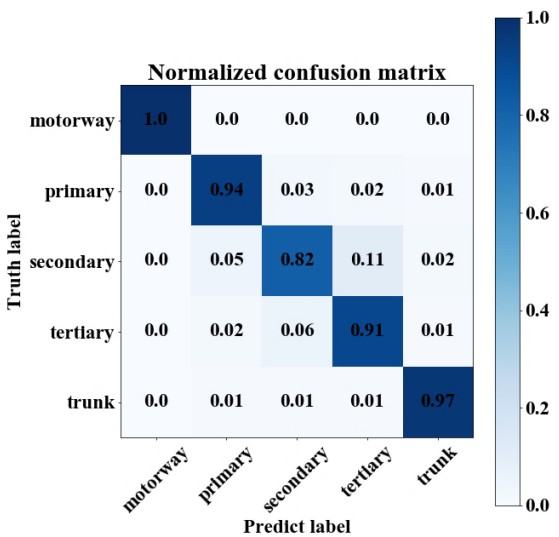

**Figure 7.** Results of MSC-DeepFM confusion matrix.

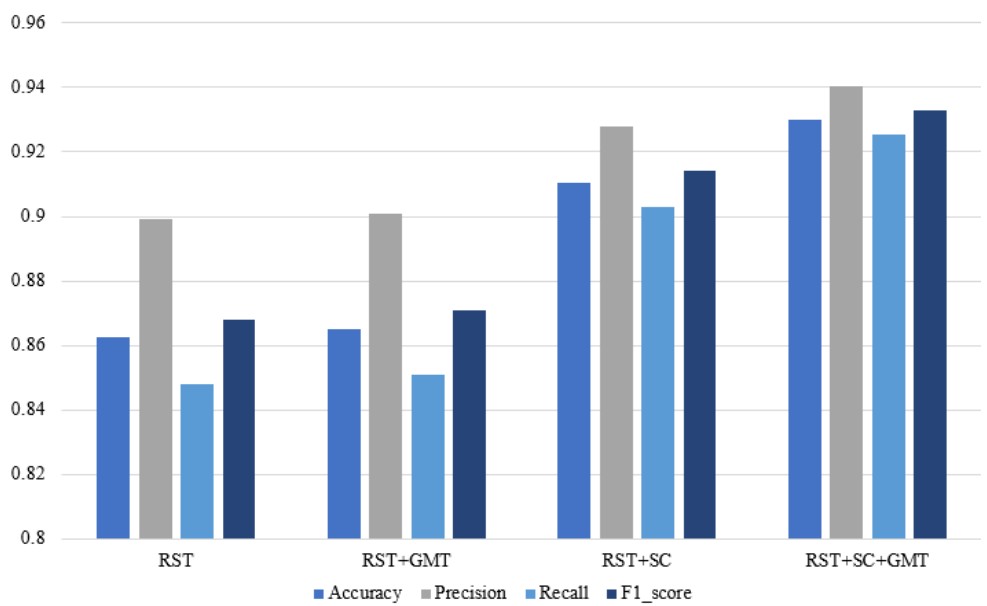

**Figure 8.** The model performance with different combinations of features. (RST, GMT, and SC represent restricted features, geometric features, and spatial context, respectively).

Spatial context influences experiments: Spatial context is the description of a geographical object about its information of surrounding objects. Those features play an important role in tag prediction through the interaction capability of the model's higher-order features. As shown in Figure 9, among the three spatial contexts, the connecting road context has the most prominent effect on the road type prediction model, which increases the F1 score to 92.17%. Similarly, both building context and POI context improve the model to different degrees. However, for connecting road context, the experiments extract the connecting road context of the object road according to the four road topologies. Since roads are interconnected to form a road network, the corresponding connecting road context can be extracted from each road, which will form a dense vector that can be significantly helpful for the training of the model.

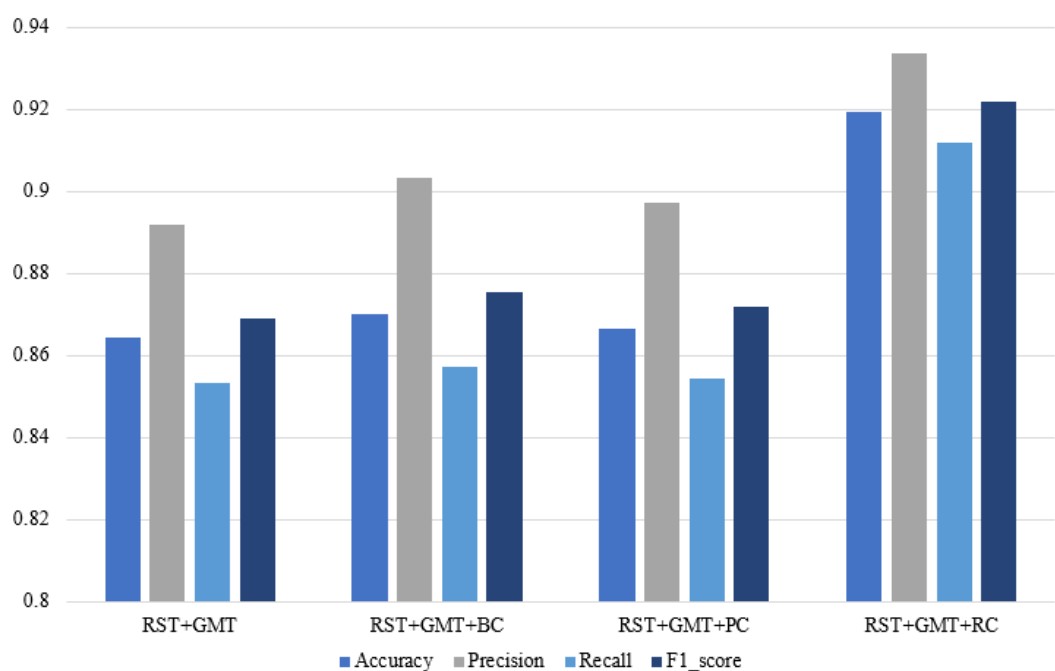

**Figure 9.** The model performance with different combinations of spatial context. (RST and GMT represent restricted features and geometric features, respectively. BC, PC, and RC represent building context, POI context, and connecting road context, respectively.

## 6. Conclusions

Current research on OSM road type prediction omits considering some useful spatial contexts and lacks interactions among features, which leads to low prediction efficiency. Therefore, we proposed a representation method of multiple spatial contexts (MSC) and a novel framework named MSC-DeepFM for OSM road type prediction. Firstly, four kinds of road topological relations are devised to model different kinds of intersections. To the best of our knowledge, we are the first one to consider road topological relations and the POI context of road objects for road type prediction. Secondly, we propose a representation method integrating multiple spatial contexts (MSC) which include the features of intersecting roads, surrounding buildings, and POIs. And the contexts are distilled to describe different road objects in OSM more accurately. Then, the multidimensional feature vectors are fed into DeepFM to enable low- and high-order feature interactions based on end-to-end learning. Finally, we designed the multi-class model of OneVsRest to complete the multi-class prediction of road types.

OSM Guangdong road data were used to verify our model. The results show that spatial context, especially connecting road context, plays an essential role in road type prediction. Compared with other machine learning methods, it shows that our MSC-DeepFM method achieves a significant improvement over others. It can solve the problem of data sparsity and semantic incompleteness. Furthermore, our model can also be used to improve the attribute completeness of OSM tag.

Our experiments mainly focus on multi-dimensional and sparse datasets in China. There are still some further verification and support through much more experiments in our subsequent studies using dense and complete OSM datasets. More road types (such as 'Residential' and 'unclassified') and more features (such as road length) will deploy in our model to verify and improve the adaptation our model. Furthermore, other open-source data, such as land cover data, satellite remote sensing data, and official data, can be used to improve the accuracy and completeness of the original OSM data.

**Author Contributions:** Conceptualization, Y.Z.; Methodology, Y.Z., Y.L., H.L. and Y.N.; Software, Y.N. and H.L.; Validation, Z.L.; Formal analysis, F.L. All authors have read and agreed to the published version of the manuscript.

**Funding:** This research was funded by the National Natural Science Foundation of China (No. 41871320), the Key Scientific Research Foundation of Hunan Provincial Education Department of China (No. 22A0341), the Hunan Provincial Natural Science Foundation of China (No. 2021JJ30276), the Science and Technology Innovation Program of Hunan Province (No. 2023SK2081) And The APC was funded by MDPI Sustainability Editorial Office.

**Data Availability Statement:** The data source for this paper is: http://download.geofabrik.de/.

**Conflicts of Interest:** The authors declare no conflict of interest.

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
