# Peer review of "MSC-DeepFM: OSM Road Type Prediction via Integrating Spatial Context Using DeepFM"

_sustainability, doi:10.3390/su152416671_

Round 1

Reviewer 1 Report

Comments and Suggestions for Authors

This study proposed an approach to predict OSM road type. The approach was achieved by integrating multiple spatial contexts with DeepFM. Results showed that the proposed model is more effective than several other methods. Generally, the manuscript is interesting and well-organized. However, several issues should be considered before it is published.

(1)  The study highlighted that the purpose is to improve the accuracy of OSM tag. But, from this study, I cannot find out how much accuracy has been improved for the study area (Guangzhou)? In my view, I prefer to define the work as ‘improve the attribute completeness of OSM tag’. Please discuss this point in the manuscript.

(2)  Page 7. In section 3.2.1.

The study used both building data and POI data to predict road type. Where is the data source of the building and POI data? It should be addressed clearly. It seems that these data were also acquired from OSM. However, to the best of our knowledge, the completeness of building and POI data are very low in China (Zhou et al. 2022). Thus, is it really effective to use the incompleteness building and POI data to predict road type? Can the approach be applied to other study areas, where the completeness of OSM building and POI data varied?

The following paper is related to this work, please cite and discuss the above point.

Zhou, Q., Zhang, Y., Chang, K., and Brovelli, M.A. (2022). Assessing OSM building completeness for almost 13,000 cities globally. International Journal of Digital Earth.

(3)  Page 7. In section 3.2.2

The authors highlighted that ‘the higher the number of junctions, the lower the road grade tends to be’. In my view, it is not always correct. This is because the number of road intersections may somehow be related to the total length of the road. That is, a longer road may have more road intersections. Further, a longer (e.g., trunk or primary) road may be more important in the network, in terms of the road type. Please consider to adjust the assumption in the manuscript.

(4)  Page 8. In section 3.2.3

The study considered five types of road attributes. They are “oneway”, “maxspeed”, “layer”, “bridge”, and “tunnel”. However, to my knowledge, the values for these attributes may not always be available in OSM. Thus, if the values are not fully available, can the proposed approach still be effective?

(5)  Page 11. In paragraph 4.

The study considered five road types for prediction. They are ‘motorway’, ‘trunk’’, ‘primary’, ‘secondary’, and ‘tertiary’. To my knowledge, there are many other road types (e.g., ‘residential’ and ‘unclassified’) in OSM. As addressed in the manuscript, the proportions of ‘residential’ and ‘unclassified’ road types are even higher (24% and 11%, respectively). Thus, can the proposed approach be used to predict more road types?

Reviewer 2 Report

Comments and Suggestions for Authors

The paper presents a compelling approach to addressing the challenges of improving the semantic quality of OpenStreetMap (OSM) data, which is a valuable resource for monitoring Sustainable Development Goals (SDG) indicators. The authors recognize the issues of data sparseness and inaccurate descriptions in OSM data and propose an innovative solution through the MSC-DeepFM model.

One of the strengths of this paper is the clear identification of the existing problems in OSM data quality, particularly in the context of road type prediction. The acknowledgment of data sparseness and inaccuracies is a crucial step in understanding the limitations of crowdsourced geospatial data. The paper's focus on improving the accuracy of road type prediction is directly aligned with the goal of enhancing the utility of OSM for monitoring SDG indicators.

The MSC-DeepFM model itself is a well-considered and comprehensive approach to address these challenges. The integration of DeepFM for handling data sparseness is a logical choice, as deep learning models have demonstrated their effectiveness in various domains. The incorporation of multiple spatial contexts, including intersecting roads, surrounding buildings, and points of interest (POIs), is a valuable addition, as it enriches the understanding of different road types. By combining these spatial contexts with geometric and restricted features within DeepFM, the model can effectively capture both low-order and high-order features, which is essential for accurate predictions.

The experimental results presented in the paper are a significant highlight. The fact that the MSC-DeepFM model outperforms some state-of-the-art methods is a testament to its efficacy. The inclusion of a multivariate classifier, OneVsRest, for road type prediction is a well-justified choice, as it allows for efficient handling of multiple classes.

However, there are areas where the paper could be improved. Providing more detailed information on the data preprocessing steps and the specific features used in the model would enhance the paper's clarity and reproducibility. Additionally, discussing the practical implications of the improved OSM data for monitoring SDG indicators and other potential applications would provide a broader context for the research.

In summary, "MSC-DeepFM: OSM road type prediction via integrating spatial context with DeepFM" presents a well-thought-out approach to address the challenges of OSM data quality for road type prediction. The integration of deep learning and multiple spatial contexts in the MSC-DeepFM model holds promise for accurate predictions. This research is a valuable contribution to the field of geospatial data analysis, and it has the potential to support more effective monitoring of SDG indicators using OSM data.

Round 2

Reviewer 1 Report

Comments and Suggestions for Authors

Thanks for responsing to all my comments! I have no further comment!